# Assessment of Changes in a Viewshed in the Western Carpathians Landscape as a Result of Reforestation

**Michał Sobala \***[ID]**, Urszula Myga-Piątek**[ID] **and Bartłomiej Szypuła**[ID]

Faculty of Natural Sciences, University of Silesia, 41200 Sosnowiec, Poland;
urszula.myga-piatek@us.edu.pl (U.M.-P.); bartlomiej.szypula@us.edu.pl (B.S.)
**\*** Correspondence: michal.sobala@us.edu.pl; Tel.: +48-32-3689-400

**Abstract:** A viewshed analysis is of great importance in mountainous areas characterized by high landscape values. The aim of this research was to determine the impact of reforestation occurring on former pasturelands on changes in the viewshed, and to quantify changes in the surface of glades. We combine a horizontal and a vertical approach to landscape analysis. The changes in non-forest areas and the viewshed from viewpoints located in glades were calculated using historical cartographic materials and a more recent Digital Elevation Model and Digital Surface Model. An analysis was conducted using a Visibility tool in ArcGIS. The non-forest areas decreased in the period 1848–2015. The viewshed in the majority of viewpoints also decreased in the period 1848–2015. In the majority of cases, the maximal viewsheds were calculated in 1879/1885 and 1933 (43.8% of the analyzed cases), whereas the minimal ones were calculated in 2015 (almost 57.5% of analyzed cases). Changes in the viewshed range from 0.2 to 23.5 km$^2$ with half the cases analyzed being no more than 1.4 km$^2$. The results indicate that forest succession on abandoned glades does not always cause a decline in the viewshed. Deforestation in neighboring areas may be another factor that has an influence on the decline.

**Keywords:** viewshed; landscape changes; reforestation; secondary succession; the Carpathians; archival maps

## 1. Introduction

Any current landscape is a result of land use and development that has changed and is changing over time [1–3]. Historical changes in a landscape also determine the future direction of its development. Research is unanimous that landscapes are evolving, and this assumption leads to the interpretation of the current state of a landscape as an effect of driving forces [4]. There are a lot of studies using cartographic retrospective analyses, e.g., changes in forest range, urban areas, rural areas, the movement of river-beds, the disappearance of lakes, changes in the occurrence of drainage ditches etc. in different spatial and time ranges. They prove quantitative and qualitative changes in a landscape [5–7]. The conducted analyses make the accurate assessment of features differentiating the structure of a landscape possible in a designated time period. Moreover, this research also has an enormous practical dimension, especially when it enables the recognition of land use conditions and an assessment of landscape sustainability [8].

This paper is devoted to an analysis of changes in the viewshed that are a result of changes in landscape structure understood as a share and proportion of different land use forms. Even though an analysis of the viewshed is not a new approach, an analysis of changes over such a long time period has not been conducted. Viewshed analysis is often used in landscape evaluation, classifying an area by its degree of visibility [9]. As far as scenic values are concerned, view ranges, number of landscape plans, horizontal and vertical layout [10], coloristic variety, taint and shape contrast and impression

analysis were also taken into account during the research [11,12]. Papers devoted to the assessment of a visual value of the landscape [13,14] and methods of visibility calculation have been developed [15–17]. Moreover, the process of openness and closure of a landscape have been analyzed in long-standing human activity, including settlement development, and the occupying and utilization of terrain for economic purposes [18–20].

A physiognomic approach to current research is also crucial due to the signing of the European Landscape Convention by the majority of European countries. In accordance with this act, "a landscape means an area, as perceived by people, whose character is the result of the action and interaction of natural and/or human factors" [21].

A viewshed analysis is of great importance in mountainous areas which are characterized by great landscape values and huge touristic potential. This results in strong touristic penetration and causes numerous threats to landscape quality, as well as its aesthetic values and sustainable utilization [22,23]. Mountainous areas make the observation of many landscape plans, remote horizons and ranges with wide views possible because of hypsometric diversity. These features are of great importance for sightseeing. Glades are places that are predisposed to these observations. However, their size and quantity are variable over time [24,25] and consequently the view range understood as a physiognomic feature of landscape has also changed. There is a strong relationship between a type of land cover (landscape openness or closure) and the visual features of a landscape. Changes in land use automatically change landscape structure, which influences physiognomic features because they depend on the viewshed. Hence, our research is an attempt to combine two different but complementary approaches in landscape analysis: a landscape as an area (horizontal approach) and a landscape as a view or scenery (vertical approach) [26]. The research task was to determine how the surface of glades and the viewshed in viewpoints located in these glades has changed. Reforestation is favorable from an ecological point of view; nevertheless, from a landscape (in particular scenic) and touristic point of view, it leads to landscape closure, visibility limitation and a decline in physiognomic attractiveness [27].

We assumed that the rate of landscape openness is a synthetic indicator of the evolution of nature–cultural environments. The process of landscape openness and closure has a fluctuational course and emerges as changes in land use proportion (mainly forest cover) caused by natural and anthropogenic factors. It was assumed that changes in landscape openness and closure were influenced by natural (reforestation and deforestation caused by long-lasting droughts, episodic winds, forest fires, pest gradations etc.) and historic–cultural factors (changes in population, political decisions, legal regulations, technical solutions and skills in land utilization). Consequently, it was assumed that the rate of landscape openness/closure should be studied simultaneously as a result of environmental transformation caused by natural factors and cultural metamorphosis [19,20,28].

The aim of this research was to determine the impact of reforestation occurring on former pasturelands on changes in the viewshed in the Western Carpathians. What is more, the aim was to quantify changes in the surface of glades and detect the causes of these changes. This is because glades are places of potentially the greatest visibility range in mountainous areas covered with forest. Research conducted in the Carpathians showed that the forest cover change was closely related to agricultural dynamics and that rates and patterns of change were heterogeneous among politically distinct time periods, and varied regionally [29]. During the 19th and early 20th century, marginal agricultural sites in the Polish mountains exhibited the most abandonment due to harsh environmental conditions [30]. Agricultural abandonment and reforestation in the Carpathians occurred mostly after WWII, with few local exceptions during earlier times [31]. We also set a methodological aim, which was the assessment of the advantages and disadvantages of several cartographical materials, presentation of its limitations and definition of the rules for conducting similar analyses.

## 2. Materials and Methods

### 2.1. Study Area

The Western Carpathians stretch from the Low Beskids range of the Eastern Carpathians along the border of Poland with Slovakia toward the Moravian region of the Czech Republic and the Austrian Weinviertel. The area of the Western Carpathians comprises about 70,000 km². The highest elevation is the Gerlachovský štít (2655 m a.s.l.). However, this mountain belt covers, besides the region of the Tatra Mountains, areas of mid- and low-mountains (the Western Beskids), foothills and valley bottoms.

The Western Carpathians have been relatively densely populated since the Middle Ages. At the turn of the 15th and 16th centuries, Vlachs shepherds came to the area of the Western Beskids, founding new settlements at higher elevations and forming glades by slashing and burning the forest for sheep to graze. For this reason, man has exerted a strong influence on land use and land cover (LULC) in the highest parts of mountain belts, leading to landscape opening. From the end of the 17th century until the mid-19th century the expansion of buildings and arable fields occurred. As a result, settlements developed on some of the glades. In the mid-19th century, mountain grazing started to collapse as a result of industrial development and the intensification of forest management connected with the Industrial Revolution. The abolition of serfdom and the stagnation in the sale of sheep products also had an influence. Since then, the surface of mountain pastures and glades has decreased [24]. Hence, this region is an excellent example for describing the problem of a decrease in viewshed, which is typical of mountainous areas in more economically developed countries [32].

We selected representative study sites for the Western Carpathians. The study area is located in the Western Beskids [33]. We carried out detailed studies in the Silesian Beskids (21 viewpoints) and in the Żywiec-Kisuce Beskids (30 viewpoints). These points were located in the mid-forest glades (Figure 1, Table 1). Each study area covers about 45 km². The glades are all located over 600 m a.s.l., which provide extensive viewpoints and determine both the attractiveness of the landscape and the attraction to tourists in the area. The availability of historical maps had an impact on the choice of study sites. The oldest preserved map, which is the Austrian cadastral maps, preserved only for selected parts of the Silesian and Żywiec-Kisuce Beskids. This, in turn, limited the spatial range of research to those places for which these maps have been preserved.

**Table 1.** Location of viewpoints.

| No | Glade | Mountain Range | Number of Viewpoints |
|----|-------|----------------|----------------------|
| 1 | Przysłop | Silesian Beskids | 4 |
| 2 | Buczyna | Silesian Beskids | 2 |
| 3 | Jaskowa | Silesian Beskids | 2 |
| 4 | Pod Skałką | Silesian Beskids | 1 |
| 5 | Ostre | Silesian Beskids | 7 |
| 6 | Radziechowska | Silesian Beskids | 4 |
| 7 | Przybędza | Silesian Beskids | 1 |
| 8 | Kikula | Żywiec-Kysuce Beskids | 2 |
| 9 | Magura | Żywiec-Kysuce Beskids | 7 |
| 10 | Bułkowa | Żywiec-Kysuce Beskids | 2 |
| 11 | Praszywka | Żywiec-Kysuce Beskids | 4 |
| 12 | Bendoszka | Żywiec-Kysuce Beskids | 3 |
| 13 | Przegibek | Żywiec-Kysuce Beskids | 4 |
| 14 | Mała Racza | Żywiec-Kysuce Beskids | 5 |
| 15 | Śrubita | Żywiec-Kysuce Beskids | 3 |

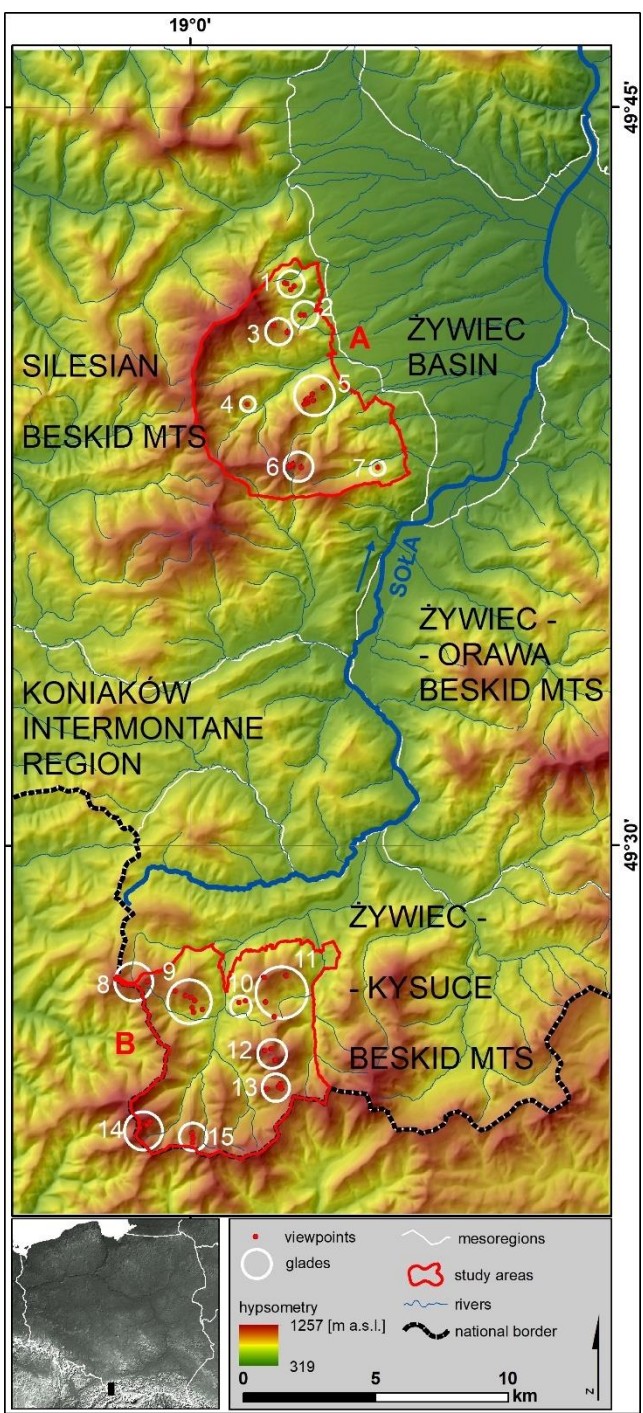

**Figure 1.** Location and hypsometry of the study area. 1–15—number of glades—see Table 1.

## 2.2. Materials

In our research, we decided to use a number of historical cartographic materials and more recent digital data (Table 2).

The oldest material that was used in the research was the Austrian cadastral map at a scale of 1:2880 [34], probably prepared in a Cassini-Soldner cylindrical transversal equal distance projection or with a non-homogenous projection based on it [35]. The next map was Spezialkarte der Österreichisch-Ungarischen Monarchie at a scale of 1:75,000 [36] drawn up in a separate location with an oblique stereographic projection [37]. The map from the beginning of the 20th century was a military

map of Poland at a scale of 1:100,000 [38] which was prepared in a quasi-geographical projection with a grid at intervals of two kilometers. The subsequent was a military topographic map at a scale of 1:25,000 [39]. This map was made in a flat rectangular coordinate system in 1942 (EPSG: 3334) based on aerial photographs [40]. The last published paper map was a topographic map of Poland [41]. It was made in a flat rectangular coordinate system in 1965 (EPSG: 2175) by the civil service for economic purposes. All these maps present the state of the environment according to their scale and year of publication. The military nature of most maps is the basis for claiming that they have been prepared with accurate precision.

**Table 2.** Cartographic materials used for the analyses.

| Map Type | Scale Resolution | Date | Map Sheet |
|---|---|---|---|
| Austrian cadastral map | 1:2880 | 1848 | Lipowa, Ostre, Radziechowska, Rycerka Górna |
| Spezialkarte der Österreichisch-Ungarischen Monarchie | 1:75,000 | 1879/1885 | Saybusch/Ujsoly-Stara Bistricca |
| Polish Military Map WIG | 1:100,000 | 1933 | Żywiec, Ujsoły |
| Military topographic map | 1:25,000 | 1960/1975 | Szczyrk, Szczyrk, Węgierska Górka/Nova Bistrica, Oscadnica, Rycerka Dolna, Zborom nad Bistricou |
| Topographic map of Poland | 1:10,000 | 1979 | Barania Góra, Lipowa, Młada Hora, Przegibek, Szczyrk Malinów, Tatarki, Węgierska Górka, Wielka Racza |
| Orthophotomap | 0.25 × 0.25 m | 2015 | - |
| Digital Elevation Model (DEM) | 1 × 1 m | 2015 | - |
| Digital Surface Model (DSM) | 1 × 1 m | 2015 | - |

The remaining data used are modern digital materials. An orthophotomap [42] with a pixel resolution of 0.25 × 0.25 m, which corresponds to a 1:10,000 scale map and is a raster, cartometric result of the orthogonal processing of aerial photographs or satellite scenes. It was also made in a flat rectangular coordinate system 1992 (EPSG: 2180). The digital elevation model (DEM) [43] and digital surface model (DSM) [44] are products based on LiDAR scanning (ALS) with a density of 4–8 points/m$^2$. They were made in a flat rectangular coordinate system 1992 (EPSG: 2180) with a 1 × 1 m resolution and an average height error in the range of 0.2 m [45].

*2.3. Methods*

The research procedure can be divided into several basic stages (Figure 2).

- In the first step, 15 glades were chosen on the basis of terrain reconnaissance and analysis of the cartographical materials. The surfaces and maximal ranges of all the glades were designated based on historical and contemporary maps for all available time periods, i.e., 1848, 1879/1885, 1933, 1960/1975, 1979 and 2015. A total of 51 viewpoints were determined for all the glades (1–7 viewpoints on each glade dependent on their surface). The location of the viewpoints was optimal, which means that they were located in the highest parts of the glades and obscured by trees in the least degree. The points were marked out in the field, which saved using a GPS receiver and then exported to shp point files. The selected study sites are located in the Western Beskids and they are representative of the Western Carpathians.
- Next, all the cartographic historical maps and modern digital spatial data were completed.
- Then, the archival sourced maps were transformed into digital versions. The archival maps were georeferenced in two steps, which comprised a calculation of the transformation matrix and

an implementation of suitable geometric transformation and interpolation and resampling of distorted images to a new standard-sized raster (i.e., so-called "rubbersheeting"). This two-step process resulted in greater georeferencing accuracy, ensuring the quality of the results and enhancing confidence in the conclusions. In each case, the georeferencing was specifically adjusted to the quality and type of data in order to achieve the best possible results for each series [46]. The Austrian cadastral maps were overlaid onto a grid whose size corresponded to that of the map frame, using affine transformation and the coordinates of the frame corners. Rectification was then carried out and its precision verified by estimating the root-mean-square error (RMSE), which was <4.91 m for each map sheet [47]. Control points were not used because of their insufficient number and their unregular distribution (this is connected with the dominance of forest cover). According to Kadaj [48], in small areas with a spatial range that does not exceed 5 km, only the transformation based on corner points can be applied. This area meets this condition. The Spezialkarte der Österreichisch-Ungarischen Monarchie map was georeferenced exclusively by means of control points of the reference layer using affine transformation [46]. This kind of georeferencing of a single map sheet yields better results than that based on fitting the corners into a millimeter mesh grid [49]. The military maps were georeferenced by overlaying the corner points of the raster image onto a grid whose size corresponded to the map frame size, using affine transformation. Rectification was then carried out and the image was adjusted to the reference layer using control points. For all maps, the historical local reference system was transformed into the contemporary global system [46]. This step involved the application of a simplified Helmert transformation with three parameters (dx, dy, dz) to a shift in the origin of the coordinate system using inverse Molodensky formulas [47].

- The fourth step was screen digitalization of the previously processed cartographic materials using the snapping function. A topology construction tool was used to detect and eliminate the errors that are usually generated during this operation, e.g., duplicated arcs, floating or short lines, overlapping lines, overshoots and undershoots, unclosed and weird polygons [50]. Screen digitalization was combined with the creation of a database of glade surfaces. By aggregating the data included in each series of maps, land-cover maps were developed in which forest and glade areas were clearly distinguishable. As a result of these procedures, vector maps were created.
- The next step was preparing all the sheets of DSM—they were mosaiced into one big raster for the whole study area. To make the viewshed analyses possible, the DSM had to be appropriately prepared: On the basis of historical maps, vector layers with forest and non-forest areas were prepared for all time moments (see fourth step above). With a layer of non-forest areas, we cut out the modern DSM [44]. Then we filled these places in DSM with data from the modern DEM [40]. In this way we prepare DSMs for each time moment.
- The last stage involved using a Visibility tool in ArcGIS [51] to conduct an analysis of visibility for each viewpoint. The following premises underpinned the calculations for all viewpoints: (a) the calculations were conducted at three different altitudes above ground level: 1.5 m, 3.0 m, 4.5 m; (b) the radius of the visibility calculation was 10 km, which comprises near (up to 1.5 km) and middle zones (1.5–10 km) distinguished by Schirpke et al. [52]; (c) the visibility analysis was conducted for time moments in accordance with the available maps.

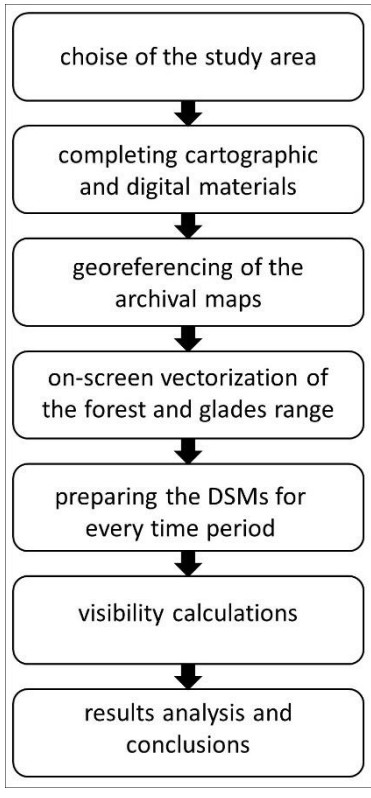

**Figure 2.** Schematic diagram of the research procedure.

## 3. Results

### 3.1. Changes in Non-Forest Areas

In both study areas in the period 1848–2015, the non-forest areas decreased systematically due to the abandonment of agriculture (Figure 3; Table 3). The decrease in non-forest areas in the Żywiec-Kysuce Beskids was higher than that observed in the Silesian Beskids (16.7% vs. 11.8%, respectively). The greatest changes in the Silesian Beskids, affecting 6.8% of the examined region, took place between 1933 and 1960, whereas in the Żywiec-Kysuce Beskids non-forest areas have changed continuously since 1933 and involve 14% of the study area.

**Table 3.** Changes in non-forest area: NFA—percentage of non-forest area, NNP—number of non-forest patches, MNA—maximal non-forest patches area.

| Study Area | Time Section | NFA (%) | NNP (ha) | MNA (ha) |
|---|---|---|---|---|
| Silesian Beskids | 1848 | 17.9 | 142 | 165.7 |
| | 1879 | 17.7 | 95 | 174.0 |
| | 1933 | 14.8 | 83 | 84.6 |
| | 1960 | 8.0 | 100 | 58.6 |
| | 1979 | 6.5 | 151 | 38.0 |
| | 2015 | 6.1 | 106 | 25.4 |
| Żywiec-Kysuce Beskids | 1848 | 31.2 | 89 | 462.3 |
| | 1885 | 30.6 | 49 | 435.8 |
| | 1933 | 28.5 | 33 | 454.9 |
| | 1975 | 21.2 | 89 | 394.5 |
| | 1979 | 20.6 | 140 | 353.7 |
| | 2015 | 14.5 | 66 | 115.5 |

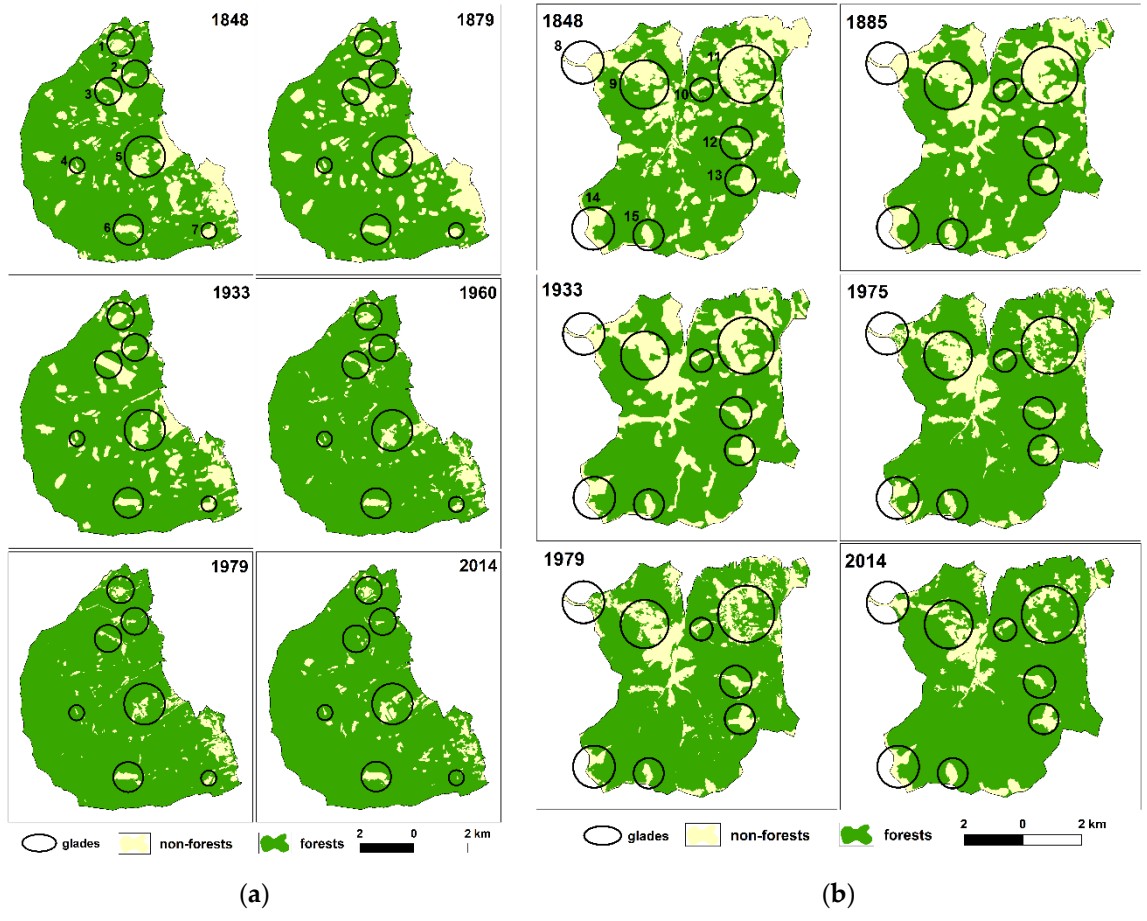

**Figure 3.** Changes in non-forest areas: (**a**) the Silesian Beskids (**b**) the Żywiec-Kysuce Beskids. 1–15—numbers of glades—see Table 1 and Figure 4.

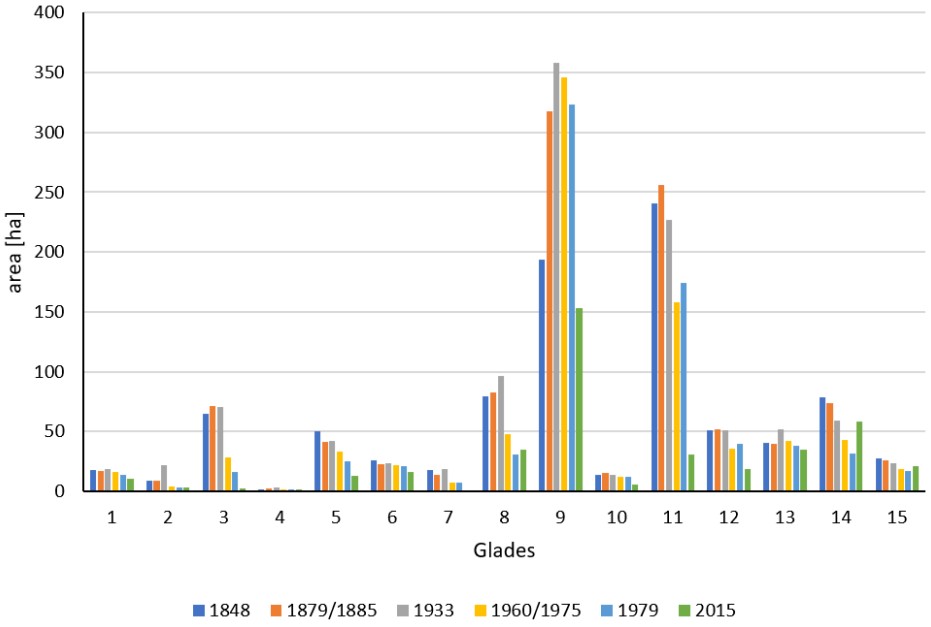

**Figure 4.** Changes in area of selected glades in the period 1848–2015. 1–15—numbers of glades—see Table 1.

The surface of all of the analyzed glades decreased in the period 1848–2015. However, the dynamics of the changes were diverse (Figure 4). The largest changes were observed on Przybędza and Jaskowa glades, where the area decreased thirty-five-fold and twenty-six-fold, respectively (from 71.5 ha to 2.7 ha and from 17.5 ha to 0.5 ha, respectively). The present-day area of Przybędza glade constitutes only 2.9% of its historical area calculated in 1848. A considerable decrease in area was also observed in Praszywka glade (almost an eightfold decrease from 256.1 ha to 30.7 ha). In turn, the largest area which was maintained was observed in the Śrubita and Mała Racza glades. As far as Śrubita is concerned, a decline of only 22% was noted. In the second half of the nineteenth century, in ten out of the fifteen glades, a small increase in area was observed. Then, since the first half of the twentieth century, a gradual decrease was noted. A small increase in area was observed in only four glades from 1975 to 1979.

### 3.2. Changes in Viewshed

The viewshed in the majority of viewpoints located in the analyzed glades decreased in the period 1848–2015 (Figure 5; Table S1). The maximal viewshed was calculated in 1879/1885 and 1933 (in both time sections, 43.8% of the analyzed cases). However, these dates were diverse for particular measurement points. The minimal viewsheds were calculated in 2015 in the majority of cases (almost 57.5% of analyzed cases). In the Żywiec-Kysuce Beskids, the minimal viewshed was observed in some cases in 1848. In some cases the viewshed increased from 1979 to 2015 and, what is more, in Radziechowska and Kikula glades, the viewshed achieved the maximal value in 2015.

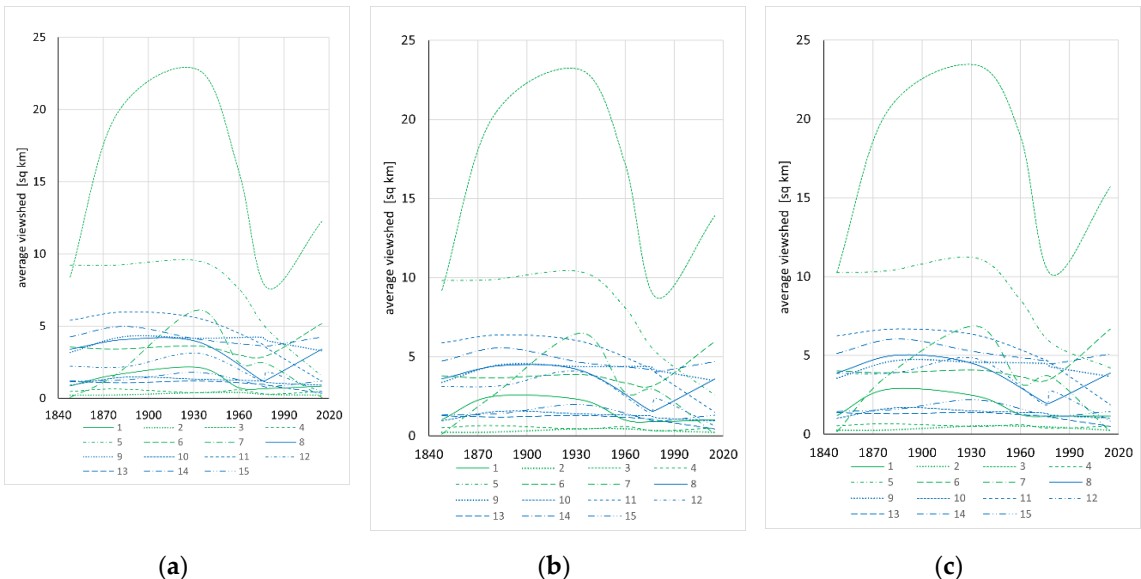

**Figure 5.** Changes in average of viewshed from viewpoints located at different altitudes above ground level: (**a**) 1.5 m; (**b**) 3.0 m; (**c**) 4.5 m. 1–15—numbers of glades—see Table 1.

Changes in the viewshed are very diverse and range from 0.2 to 23.5 km$^2$ for particular viewpoints with half the cases analyzed being no more than 1.4 km$^2$. As the altitude above ground level increases, these values do not change much. Only a small decrease was observed in the maximal differences between the highest and lowest viewshed as the altitudes measured are higher (these differences are as follows: 23.5 km$^2$ for points located 1.5m above ground level, 22.3 km$^2$ for 3 m and 20.6 km$^2$ for 4.5 m). The average changes in viewshed from all viewpoints for all altitudes above ground level in the analyzed time period are presented in Figure 6.

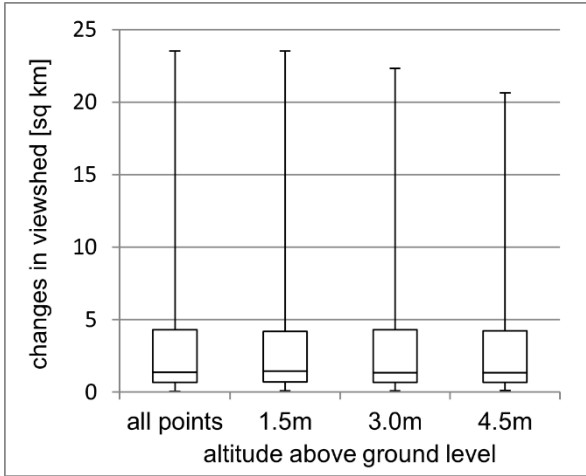

**Figure 6.** Distribution of amounts of maximal changes in viewshed in the period 1848–2015.

## 4. Discussion

In this article we combine two different but complementary approaches in landscape analysis: a landscape as an area (horizontal approach) and a landscape as a view or scenery (vertical approach). To this end, we quantify changes in the surface of glades and in viewsheds from viewpoints located in these glades. The results show that the non-forest areas decreased systematically in the period 1848–2015 in all of the analyzed glades. Moreover, the viewshed in the majority of viewpoints located in these glades decreased in the same time period. However, the dynamics of the changes in non-forest areas and viewsheds were diverse.

The analysis of changes in glade areas indicates that the process of secondary succession on the abandoned glades is progressing. The increase in forest range in the study area is connected with land abandonment [6,53]. Reforestation is typical of many mountainous areas in economically developed countries [54–56]. This process is accompanied by secondary succession of forest, which results in wide obscuring of panoramas and a decline in viewshed. Until now, the reasons for, dynamics of, and natural and economic results of forest transition have been widely analyzed [57,58]. The method proposed in this paper might be useful to identify areas with wide changes in viewshed, which can represent a priority for landscape protection policies.

The intensity, duration and beginning of reforestation has an influence on the decrease in viewshed [59]. This process has been progressing in the study area since the turn of the nineteenth and twentieth centuries. The analysis of archival maps indicated that this area was deforested to the maximal extent in the second half of the nineteenth century. The landscape was characterized by great diversity, which was connected with the diversification of land use. This is typical of the majority of European landscapes at that point [60,61]. However, in the Carpathians considered as the entire region, forest cover decrease stopped between WW I and WW II. It indicates that the Carpathians experienced a forest transition during the Interwar period, despite regional differences [29,62]. The beginning of the decline in viewshed in the study area at the turn of the nineteenth and twentieth centuries should be associated with the development of forest management carried out by the then owners of forests in the Beskids—The House of Habsburg—who introduced restrictions on the use of forests by the local population, and reforested the previously purchased glades [63]. Although the total forest cover increased in the study area, the area of forests with a long temporal continuity (potential aging and potential old-growth forest) declined what is typical of other regions of the Carpathians [64].

Agricultural abandonment in the study area started earlier than in the whole Carpathians, where it was a prominent process during the Interwar and Socialist periods [29]. It might be due to harsh environmental conditions and it caused decrease in forest fragmentation [30,65]. Research in different parts of the Carpathians highlighted regional variation in land change patterns, and in

the major drivers of change. Reforestation continued after WW II because many glades were not used for grazing [46]. However, abandonment increased after the collapse of the Socialism. It was also connected with the political and socio-economic transformation at the turn of the 1980s and 1990s which took place in Central and Eastern Europe, and which accelerated land abandonment in upper mountain locations [66–68]. The main reasons were the lack of agricultural subsidies, decreased profitability, availability of better-paying jobs outside agricultural sectors and rural–urban migration [69,70]. This emphasizes the importance of socioeconomic factors for farmers' decisions related to the cessation of land cultivation [71].

However, in recent years, there has been a slowdown in the decrease in viewsheds, and in some cases this range has also increased. First of all, the slowdown in the decrease may be connected with the course of secondary succession. Ciurzycki [72], using the example of the glades of the Tatra Mountains, pointed out that, in subsequent stages, overgrowth of glades progresses through the growth of existing trees, and not by the appearing of a new generation of trees. This may be connected with the diversity of light, heat, humidity and fertility conditions and competition of natural vegetation preventing the encroachment of further trees; in this way, the process of succession is probably cramped. In turn, the increase in viewsheds may be connected with the catastrophic results of improper forestry beginning in the nineteenth century. At that point, the natural beach–fir forests compatible with habitat were displaced by spruce monocultures that were preferred by forest management for lumber manufacturing. As a result, the process of tree dieback has been observed since the 1970s, which has consequently led to the clear-cutting of snags on large areas of hillsides. Subsequent to this, wide panoramas were revealed temporarily, not only within glades but also up to the present forested areas [46].

Therefore, not only does the course of secondary succession of glades have an influence on the viewshed from particular viewpoints located in glades, but also the changes in forest range in neighboring areas. Furthermore, in both cases, the effects of nature and landscape conservation are visible. They are conducted by nature protection services as well as by local associations that aim to restore traditional sheep grazing, mowing of meadows and cutting out of single trees. As a result, secondary succession is hindered. These methods are commonly used as a way to protect landscape and biodiversity in mountainous areas where traditional grazing was typical [73,74]. These activities have been conducted in Radziechowska glade for a few years. In the neighborhood of this glade, forest clear-cutting was also observed recently as a result of spruce dieback [46]. Both of these factors recently led to an increase in visibility range.

While for the entire Carpathians area, as well as other mountainous areas in economically developed countries, the increase in forest area and the accompanying decrease in the viewshed are typical, some locally based lapses can be found [75]. Despite the impact of the same political and socioeconomic factors, regional differences in land use change were also observed by Munteanu et al. [29]. They compared 102 case studies from six countries located in the Carpathian Mountains. In turn, MacDonald et al. [76] compared land use changes in different mountainous regions of Europe and, as a result, noticed that these changes can take different forms (conversion of meadows to pastures, reduction of grazing in the highest parts of the mountains, modernization of animal husbandry, land abandonment and intentional reforestation). Hence, the dynamics of secondary succession and landscape closure will be different in particular areas, thereby influencing the dynamics and course of changes in view values. The glades where a permanent settlement exists (Przysłop glade), land abandonment has recently started, and sheep herding was restored countering spontaneous forest succession (Mała Racza and Radziechowska glade) are characterized by the most durability.

Archival maps are the only resource that makes an analysis of changes in non-forest areas and viewsheds in the past possible. Their quality has a great influence on the credibility of the results, which may be burdened with errors. For this reason, the oldest map created in the period 1779–1783, based on the first military survey of Galicia (Josephine land survey of Galicia, Originalaufnahme des Königreiches Galizien und Lodomerien) at a scale of 1:28,000, was not used. This is because of the

fact that this map is not based on a geodetic control network and does not meet the requirements of cartometry, which is understood as a map feature meaning that it may be used to carry out measurements to determine the quantitative features of the phenomena presented. In addition, materials that would clearly define the mathematical formulas of map projection did not remain, and the maximum error in the location of objects exceeds 1 km [77].

The maps used in this study differ both in terms of scale (from 1:2880 in the Austrian cadastral maps to 1:100,000 in the Polish Military Map), their use (military or administrative purposes) and map projection. Thereby, the depth and accuracy of the analysis is diverse due to the use of several maps, and the results based on these maps need careful interpretation and verification using other data sources. However, it must be emphasized that spatial detail of Polish Military Map WIG with the scale 1:100,000 is comparable with maps in much higher scale [78]. Furthermore, the results could be affected by errors occurring at each stage of the creation of a digital map, particularly georeferencing, which greatly affects the quality of results [79]. Furthermore, the value of information from map data is lower than from the direct source data, having a lower precision and accuracy. Being aware of the limitations of maps is the basis for drawing correct conclusions [80]. For example, the small changes in the area of glades in the Żywiec-Kysuce Beskids from 1975 to 1979 may be a result not of real changes, but of the difference in scale of both maps (1:25,000 and 1:10,000, respectively).

Limitations in this type of analysis are also connected with the content of particular maps. Old maps can only provide partial information for landscapes because many facts and processes cannot be depicted on maps [81]. Other sources (mostly written sources of archival information) may contribute highly accurate information to the data gathered from old maps, but these sources present serious disadvantages. For example, the scant historical information that is available is largely anecdotal and narrative and is not available at appropriate spatial and temporal scales. This complicates the process of drawing comparisons across time and space [82]. It is only possible to conduct the method proposed in this paper using maps. It must be emphasized that the range of temporal forest clearings in the study area may only be assessed based on contemporary maps. The archival maps did not contain such information; hence, changes in viewsheds from glades do not take temporary uncovering of panoramas connected with logging into account. On the other hand, this situation allows an assessment of the influence of forest succession occurring on glades on changes in viewshed eliminating the influence of cutting off the forest in neighboring areas. This activity is conducted with forest management and causes only temporary changes in forest range. Furthermore, it must be emphasized that these analyses are limited only to time periods that are presented on maps. Changes occurring between these time periods may only be interpreted based on written sources showing particular directions of changes in land use in particular time periods. The time range that can be analyzed in particular regions is different and depends on the availability of credible archival maps [83].

It must be emphasized that there are some limitations to the usefulness of the method proposed. However, they open new research prospects. Viewshed analysis is often used in landscape evaluation, classifying an area by its degree of visibility. This approach may, however, be unable to consider the value of landscape features that can be observed, as it is only based on terrain morphology features [84]. Viewshed quality is also dependent on other landscape features, such as buildings or vegetation. Their position can significantly affect human perception, as even slight changes in these elements may alter the visibility conditions [85]. What is more, from a landscape protection perspective, it is important to distinguish not only which areas are visible but also what is visible. However, the study area is rather homogenous as the majority of it is covered with forest. Other methodological improvements may also include the modelling of atmospheric and environmental conditions, as they can reduce visibility and viewshed extent [9]. Changes in viewshed may be also determined using cumulative viewshed analysis [86]. However, our aim was to check how the viewshed from individual viewpoints has changed from the position of an observer–tourist. This is important for this area because it is attractive for tourists and covered with a relatively dense network of walking trails [87].

## 5. Conclusions

The conducted research allows the following conclusions to be drawn:

- The area of all of the analyzed glades decreased in the period 1848–2015; however, the dynamics of these changes were different and were influenced by local conditions.
- The viewshed from the majority of viewpoints located in the analyzed glades decreased in the period 1848–2015. However, the direction and dynamics of these changes in the analyzed time period were different and do not always refer directly to changes in the area of the glades.
- The results indicate that forest succession on abandoned glades does not always cause a decline in viewshed. Deforestation in neighboring areas may be another factor that had an influence on the decline which was observed in some glades in 2015.
- The choice of viewpoints was arbitrary but was argued from the point of terrain awareness and analysis of maps: orthophotomap and DEM. As a result, the optimal locations were chosen (i.e., the highest altitude in glades and that obscured by trees in the lowest degree). However, the analyses of the viewsheds are objective due to their quantitative character.
- The analyses of changes in viewshed for a longer period of time may only be conducted using archival maps. This method has a limitation connected with these maps: their content, accuracy and the fact that they present only the state of the environment according to the map's publication date.
- The proposed method of viewshed analysis based on archival maps could fortify numerous research studies that analyze landscape transformation through the prism of land use changes (a landscape as an area) with an approach concerning changes in landscape aesthetics (a landscape as a scenery) using very high resolution digital surface models.

**Supplementary Materials:** The following are available online at http://www.mdpi.com/2073-445X/9/11/430/s1, Table S1: Changes in visibility range in the period 1848–2015.

**Author Contributions:** M.S. and U.M.-P. wrote the main manuscript. M.S. and B.S. conducted GIS analysis. All authors have read and agreed to the published version of the manuscript.

**Funding:** This research received no external funding.

**Acknowledgments:** We gratefully acknowledge two anonymous reviewers for their constructive comments.

**Conflicts of Interest:** The authors declare no conflict of interest.

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
