# Peer review of "Assessment of Changes in a Viewshed in the Western Carpathians Landscape as a Result of Reforestation"

_land, doi:10.3390/land9110430_

Round 1

Reviewer 1 Report

Review of the paper:  

Assessment of changes in a viewshed in the Western Carpathians landscape as a result of reforestation

The paper deals with an interesting topic related to changes in the Western Carpathians.

The reviewer recommends major revisions before publication.

  1. To discuss in the Introduction more about the changes in the Carpathians, see Munteanu et. al. https://www.sciencedirect.com/science/article/abs/pii/S0264837714000131
  2. Discuss forest continuity in Carpathian https://link.springer.com/article/10.1007/s10531-013-0523-3
  3. To addresses two research questions more specifically. To refer more clearly to changes and driving forces. In the present form, the aim is too general.
  4. It is not understood what means the vertical and horizontal approach? Vertical means temporal analysis, Horizontal means spatial analysis? Provide citations.
  5. There are many details related to the historical maps, but it is not understood how the data were extracted to complete the study? Is it just a visual evaluation of the maps or was a temporal analysis done? There are no maps to show the evolution of the authors, see: https://www.sciencedirect.com/science/article/abs/pii/S0143622815001800

6. Section 2.2: the authors must clearly explain what elements were extracted from the map of how the overlap was done from one year to another. Especially since there are big differences in scale, projection, degree of detail of the map.
The technical details about the historical maps must be put in the table and highlighted only what is important.

The article has the potential to be published in the Land, but only after a complete revision. In this phase, it is unclear and incomplete.

The authors have to provide more details about changes in Carpathian, see also MDPI journals (Sustainability and Land).

Reviewer 2 Report

I believe that the analysis is too subjective for some aspects, for example for the choice of Viewpoints.

A landscape changes analysis should be carried out by assessing the contribution of these surfaces over the whole study area, not only from the points of view .

The use of cumulative viewshed analysis is certainly more appropriate than traditional viewshed analysis for landscape works.

Round 2

Reviewer 1 Report

The manuscript has been improved. I accept this form.

Reviewer 2 Report

The changes can be accepted.